# Periodontal Health and Use of Oral Health Services: A Comparison of Germans and Two Migrant Groups

**DOI:** 10.3390/ijerph16163000

**Published:** 2019-08-20

**Authors:** Daniel Hagenfeld, Heiko Zimmermann, Katja Korb, Nihad El-Sayed, Julia Fricke, Karin Halina Greiser, Jan Kühnisch, Jakob Linseisen, Christa Meisinger, Marc Schmitter, Ti-Sun Kim, Heiko Becher

**Affiliations:** 1Section of Periodontology, Department of Conservative Dentistry, University of Heidelberg, 69120 Heidelberg, Germany; 2Department of Periodontology and Restorative Dentistry, University-Hospital of Münster, 48149 Münster, Germany; 3Institute of Global Health, University of Heidelberg, 69120 Heidelberg, Germany; 4Department of Orthodontics, University of Heidelberg, 69120 Heidelberg, Germany; 5Division of Cancer Epidemiology, German Cancer Research Center, 69120 Heidelberg, Germany; 6Institute for Social Medicine, Epidemiology and Health Economics, Charité Universitätsmedizin Berlin, 10117 Berlin, Germany; 7Department of Conservative Dentistry and Periodontology, Ludwig-Maximilians-University of Munich, 80336 München, Germany; 8German Research Center for Environmental Health, Institute of Epidemiology II, Helmholtz Zentrum München, D-85764 Neuherberg, Germany; 9Department of Prosthodontics, University of Würzburg, 97070 Würzburg, Germany; 10Institute for Medical Biometry and Epidemiology, University Medical Center Hamburg-Eppendorf, 20246 Hamburg, Germany

**Keywords:** periodontitis, migrants, oral health, dental health care

## Abstract

A cross-sectional study was performed with 251 individuals, consisting of 127 Germans, 68 migrants from Turkey, and 56 resettlers (migrants from the former Soviet Union with German ancestors) to compare periodontal health status, with a special focus on associations with lifestyle and anthropometric factors, and use of dental health services. Maximal pocket depth was used as a clinical surrogate marker for periodontitis. Other variables were obtained by questionnaires administered by a Turkish or Russian interpreter. The age- and sex-adjusted prevalence of periodontitis was significantly higher in Turks (odds ratio (OR) 2.84, 95% CI = 1.53–5.26) and slightly higher in resettlers (OR = 1.33, 95% CI = 0.71–2.49). These differences are partly explained by a differential distribution of known risk factors for periodontitis. A full model showed a higher prevalence of maximal pocket depth above 5 mm in Turks (OR = 1.97, 95% CI = 0.99–3.92). Use of oral health services was significantly lower in the two migrant groups. Individuals who reported regular visits to a dentist had significantly less periodontitis, independent of migrant status. A reasonable conclusion is that, since oral health causes major chronic diseases and has a major effect on total health system expenditures, public health efforts both generally and specifically focused on migrant groups are warranted.

## 1. Introduction

Migration occurs due to economic situations and changes in urbanization worldwide [1]. Socio-cultural determinants show further differences between countries and especially between ethnic groups. Only a few studies have included different ethnic groups to study the impacts of migration and ethnicity on oral diseases, such as periodontitis and gingivitis [2,3] or caries [4]. Chronic oral infections have been shown to be associated with severe diseases, such as strokes [5] and heart diseases [6,7]. Oral health therefore is a central public health issue, especially as the population is aging and oral health-related diseases increase, and their treatment in early stages is getting more and more important. The prevalence of periodontitis varies within countries, in urban and rural areas [8] or between lower and higher socioeconomic groups [9], and also by age, sex, smoking, glycated hemoglobin, and ethnicity [10]. Differences in income and education between countries become apparent when looking at trends in socioeconomic inequalities [11]. In particular, income and educational levels differ between ethnic groups [12,13]. Different educational levels impact the individual job situation of persons, and educational levels and degrees vary between countries of origin [11,14].

In Germany, the biggest migrant group is about 1.5 million Turkish people [15]. There are also more than 3.2 million German citizens that are resettlers from the former Soviet Union [16]. Comparisons of large dental surveys showed that, in the United States, ethnicity is included in large epidemiological surveys, but this is not true for those epidemiologic studies conducted in Germany [17,18]. One study considered a mixed migrant group in Germany and found that elderly migrants had poorer oral health status and a lower utilization of dental health services [19]. Therefore, it can be hypothesized that ethnicity and migration itself are not a risk factor for periodontitis alone, but can be explained by an accumulation of other risk factors or the poorer utilization of health services.

This paper has the following objectives: (1) To compare the prevalence of periodontitis in Germans, Turks, and resettlers, (2) to assess whether prevalence differences can be explained by known risk factors for periodontitis and some general lifestyle factors which are known to be differently distributed in these groups, and (3) to compare the use of oral health services.

## 2. Study Population and Methods

This is a cross-sectional study combining two feasibility studies within the pretest phase of a population-based epidemiologic cohort study called the German National Cohort (NAKO) [20]. In one of these studies, the feasibility of including migrants was investigated [21], and in the other, we tested procedures for oral health examinations [22]. Two of the 18 participating centers in the NAKO took part in both of these two feasibility studies, Heidelberg (southwest Germany) and Augsburg (southern Germany). We considered Germans and individuals from two different migrant groups, individuals with a Turkish background, and resettlers from the former Soviet Union with German ancestors, who participated in the oral examination program. The selection of these migrant groups is described in detail in another publication by Reiss et al. [21]. Briefly, we used two approaches. First, from the local registration offices, random population samples were drawn. The identification of persons of Turkish origin and ethnic German immigrants from the former Soviet Union in the registries was performed via nationality, country of birth, or by using an algorithm based on name. Second, we used a community-orientated approach to recruit individuals from the two migrant groups. The German group was selected from the random population sample.

The joint overall sample incorporated 251 individuals (127 Germans, 68 individuals with a Turkish background, and 56 resettlers). The study was approved by the ethics committees of the Medizinische Fakultät der Universität Heidelberg and the Bayerische Landesärztekammer, München.

All participants gave written informed consent. Individuals were invited by letter, with a written reminder, and, if required, additional contact by phone. Each person was interviewed with a standardized questionnaire that included lifestyle and social factors. They underwent a full program of medical examinations, which included a blood sample and the oral examination as described below.

The periodontal examination was conducted by study nurses supported by experienced dentists. Study nurses received a two-week intensive training program and calibration for these examinations. A total of 250 individuals were examined, both by study nurses and dentists. The agreement between both was good: ~95% agreement regarding pocket probing depths between study nurses and dentists on examined sites (*N* = 6125 out of 6394), with an error range of ±2 mm.

Maximal pocket depth (maxPD) was used as one indicator for the presence of periodontal inflammation. In the present study, the periodontal assesment had to be limited to the maximal pocket depth due to time constraints of the dental examination regiments. Therefore, the pocket depth was used as a main indicator, knowing well that the clinical diagnosis of periodontitis had to consider several other clinical indicators, e.g., bleeding on probing. A full-mouth registration for the periodontal status was conducted in Heidelberg, and a half-mouth registration was carried out in Augsburg. The maxPD was measured on at least two sites per tooth (mesial-buccal and buccal) on the maxillary and mandible portions. For the examination, a UNC-PCP15 Color-Coded Probe (Hu-Friedy Europe, Rotterdam, The Netherlands) with a black band for each millimeter, up to 15 millimeters, was used. According to a simplified staging protocol [7] only considering the maxPD, the following definition for periodontitis was used for stage (I): Pocket depth 0–3 mm as no/mild periodontitis, stage (II): At least one pocket ≥4 mm and <6 mm as moderate, and stage III: At least one pocket ≥6 mm as severe periodontitis.

Frequency of visits to a dentist was assessed with the question: “When was the last time you visited your dentist?” Answers were classified into two groups: Within the last 12 months or more than 12 months ago.

## 3. Statistical Methods

In all analyses, age was considered in 10-year age groups (≤29, 30–39, 40–49, 50–59, ≥60 years). Body mass index (BMI) was included as a continuous variable. Physical activity was considered as the metabolic equivalent of task (MET) as MET minutes/week in two groups (≤1,500, >1,500). Alcohol consumption was dichotomized into <2 times/week or never and ≥2 times/week. Smokers were classified as never smokers, ex-smokers, and current smokers. Household income was considered as <2000 €/month and ≥2000 €/month. Education was classified as years in school (up to 12 years, 13 years, and more).

The prevalence of periodontitis, dentist visits, and covariables was analyzed descriptively with appropriate frequency tables. The association of variables with main target parameters was analyzed with ordinal and standard logistic regression. The dependent variable was coded as follows: Pocket depth: ≤3 mm: 0; 4–5 mm: 1; 6+ mm: 2, and dentist visits: Once per year or more often: 0; less than once a year: 1. Since we were interested in differences between the migrant groups, and to which degree possible differences can be explained by lifestyle factors, which may be differently distributed in these three groups, we built a model with the factor “migrant group” only, and a model in which potential confounders were included. Statistical calculations were performed using SAS version 9.3 (SAS Institute, Cary, NC, USA).

## 4. Results

The overall sample consisted of 251 persons (56.6% females and 43.4% males). Age ranged from 18 years to 70 years, with Turks being significantly younger than Germans and resettlers. The proportion of individuals in the German group was about twice as high as that of the migrants. Table 1 gives the distribution of center, sex, and age variables by ethnic group. A table with the distribution of relevant covariables is given in the Appendix A.

The distribution of the indicator for periodontitis and the frequency of dentist visits by ethnic group are presented in Table 2. The number of persons with severe periodontitis is high, at around 25%, and showed little difference between sexes. The age-adjusted prevalence of severe periodontitis is higher, with odds ratios of 2.84 and 1.33 in Turks and resettlers, respectively, compared to Germans. Overall, 41.4% of persons had a maximal pocket depth less than 4 mm.

Table 3 shows the results of the crude and full multivariable regression model for maximal pocket depth and last dentist visit. The prevalence odds ratios for the migrant groups are reduced after adjusting, indicating that part of this relationship can be attributed to the other covariables. The strongest association with maxPD was found for current smokers vs. never-smokers (*p* < 0.001). Furthermore, last dentist visit longer than 12 months (*p* = 0.049) and higher alcohol consumption (*p* = 0.04) were significantly associated with maxPD. Body mass index (BMI), physical activity, educational level, and household income showed no significant association. However, the adjustment for all factors reduced the observed univariate association of ethnic groups considerably. Resettlers showed a similar prevalence of maxPD if adjusted for the above covariables. Turks still showed a higher, yet borderline, significant association for maxPD compared to Germans, indicating that other covariables that are more prevalent in Turks may play a role here. There was no influence of the study center on maxPD.

We further analyzed factors that may have an association with frequency of dentist visits, measured as last visit more or less than 12 months ago. Turks and resettlers reported lower frequency (odds ratio (OR) = 3.35, 95% CI = 1.51–7.44 and OR = 5.24, 95% CI = 2.33–11.76, respectively) compared to Germans. Frequency of dentist visits also depended on age. This effect is independent of other covariables. Except for male sex having higher odds for later dentist visits, no other variables showed a relevant effect. Interestingly income does not seem to have an effect.

## 5. Discussion

This study indicates that the prevalence of periodontitis indicated by maximally deep pocket depth was significantly higher in the two migrant groups, and these differences can partly be explained by known risk factors for periodontitis. Use of oral health services was significantly lower in both migrant groups. However, for migrants with a Turkish background, other variables seem to play a role, since a difference remained after adjustment.

Our study participants covered the age range 18–70 years, with a mean age of about 45 years. Age is known as the strongest risk factor for periodontitis, and we also found a clear trend in our study. Among the other factors, current smoking was the only clear risk factor for higher pocket depth compared to non-smokers. Similar associations with smoking have been reported earlier [22,23]. There was a significant association with sex. Males revealed a higher proportion of higher pocket depth and infrequent dentist visits than females.

To our best knowledge, no representative data on the comparison between those different ethnic groups in Germany based on periodontal outcomes are available in the literature. Kassebaum et al. [24] showed a significant decline in prevalence and incidence of severe tooth loss between the years 1990 and 2010 at the global (inter alia in Asia-Pacific, Central Europe, and Western Europe) and country levels (such as Turkey). Another survey detected a higher prevalence of periodontitis in the German population when comparing those with two other groups in Japan [25].

However, a positive non-significant effect of educational levels has been observed, in line with results from earlier studies [26,27]. Additionally, it has been shown that low parental socioeconomic status [28] and a neighborhood with low socioeconomic status [29] increase risks for periodontitis up to twofold. For different ethnic groups, especially for African-Americans and individuals with low education and income, a higher prevalence of periodontal diseases has been shown [12,13,30].

We assessed periodontal disease by the maximal pocket depth as a modified form of the well-known community periodontal index of treatment needs [31], commonly used in large epidemiological studies in the United States [32] and also in Germany [7,33]. However, non-periodontitis associated reasons for bone loss and/or local gingival enlargement might also be a reason for deep pocketing, and therefore our disease estimates might be increased. We acknowledge that the full clinical diagnosis of periodontitis has to consider several other indicators, such as bleeding on probing. This was not possible due to time constraints; however, we think that the resulting misclassification is small. 

This study has a number of shortcomings. Due to language barriers, the time for the examination of migrants was higher than for the native German population, and although efforts were made to clearly explain the questions, we cannot completely rule out resulting misunderstandings in variables. To minimize problems in respect to language, bilingual staff members were included to contact persons during planning and recruiting. Furthermore, bilingual interviewers were also present during the examination procedures. Due to the examination time limits of this study, variables for gingival inflammation, such as bleeding on probing and other potentially important variables, such as psychosocial stress, systemic diseases, and blood values, were not examined. Including the latter might further explain some of the found associations. Response rates of migrants were lower than in the native population, and as an additional scheme, recruitment via networks was used. This group cannot be considered as a random sample of the population. We cannot exclude the possibility that both the low response rate and the recruitment had an effect on the representativeness with respect to oral health. However, we could assume that those who participated had better oral health and a higher use of oral health services, so that the difference with the German group is rather under- than overestimated, and we therefore consider our conclusions justified.

In summary, the study contributes to the knowledge on oral disease prevalence and frequency of oral health service use in different population groups in Germany, and confirmed the effect of known risk factors on periodontitis. It also showed higher prevalence of periodontitis in the two largest migrant groups in Germany, the resettlers, and the Turkish migrants, which goes hand in hand with lower health service use in these groups.

## Figures and Tables

**Table 1 ijerph-16-03000-t001:** Center, sex, and age distribution by ethnic groups.

Center		Ethnic Group			
		Resettlers	Turks	Germans	Total
Augsburg	total (m,f)	7 (2, 5)	2 (1, 1)	36 (16, 20)	45 (19, 26)
	Age ($\bar{X}$ (sd))	56.7 (12.5)	43.0 (4.3)	55.7 (11.3)	55.3 (11.4)
Heidelberg	total (m,f)	49 (17, 32)	66 (30, 36)	91 (43, 48)	206 (90, 116)
	Age ($\bar{X}$ (sd))	44.5 (13.0)	39.5 (12.6)	42.6 (15.1)	42.1 (13.9)
both	total (m,f)	56 (19, 37)	68 (31, 37)	127 (59, 68)	251 (109, 142)
	Age ($\bar{X}$ (sd))	46.1 (13.5)	39.6 (12.4)	46.3 (15.3)	44.4 (14.4)

Male (m); female (f); mean ($\bar{X}$); standard deviation (sd).

**Table 2 ijerph-16-03000-t002:** Overall distribution of maximal pocket depth (maxPD) and frequency of dentist visits, by ethnic group and sex.

Ethnic Group		maxPD			Last Dentist Visit *		Total
		≤3 mm	4 + 5 mm	≥6 mm	Less than 12 Month Ago	More than 12 Months Ago	
Resettlers	total (m, f) %	24 (3, 21) 42.8%	16 (6, 10) 28.6%	16 (10, 6) 28.6%	36 (10, 26) 64.3%	20 (9, 11) 35.7%	56 (19, 37) 100%
Turks	total (m, f) %	22 (5, 17) 32.3%	28 (16, 12) 42.2%	18 (10, 8) 26.5%	48 (20, 28) 70.6%	20 (11, 9) 29.4%	68 (31, 37) 100%
Germans	total (m, f) %	58 (29, 29) 45.7%	40 (17, 23) 31.5%	29 (13,16) 22.8%	111 (48, 63) 88.1%	15 (11, 4) 11.9%	127 (59, 68 *) 100%

* One German woman with missing value for Last Dentist Visit.

**Table 3 ijerph-16-03000-t003:** Results of multivariable ordinal logistic regressions on maxPD and Last Dentist Visit by ethnic groups.

Effect	maxPD (mm), *n* = 251						Last Dentist Visit (Y), *n* = 250 *					
	Crude Model with Ethnic Group, Only ^§^			Fully Adjusted Model ^§^			Crude Model with Ethnic Group, Only ^§^			Fully Adjusted Model ^§^		
	OR	95% CI	P	OR	95% CI	P	OR	95% CI	P	OR	95% CI	P
**Ethnic group**												
Germans	1			1			1			1		
Resettlers	1.33	0.71–2.49	0.37	0.94	0.46–1.95	0.87	5.24	2.33–11.8	**<0.01**	5.13	2.07–12.75	**<0.01**
Turks	2.84	1.53–5.26	**<0.01**	1.82	0.91–3.62	0.09	3.35	1.51–7.44	**<0.01**	2.73	1.14–6.54	**0.02**
**Lifestyle and anthropometric factors**												
BMI (increase by 5 kg/m²)				1.12	0.84–1.48	0.45				0.77	0.53–1.13	0.19
Alcohol (more or equal vs. less than 2 times per week)				0.50	0.25–0.98	**0.04**				0.62	0.25–1.54	0.30
Physical Activity (high vs. low)				0.96	0.55–1.66	0.87				1.10	0.54–2.27	0.79
Smoking Never				1						1		
Ex-smoker				1.16	0.60–2.25	0.66				0.74	0.29–1.88	0.53
Current				3.65	1.91–6.98	**<0.01**				1.62	0.75–3.49	0.22
**Socio-economic factors**												
Household income per month ≥ 2000€ vs. <2000€				0.94	0.54–1.60	0.81				0.89	0.45–1.73	0.73
School education (13 years vs. <13 years)				0.61	0.34–1.10	0.10				0.76	0.35–1.63	0.47
**Last visit to a dentist (<12 vs. ≥12 month ago)**				1.93	1.00–3.69	**0.049**				-	-	-
**Sex (m vs. f)**				1.99	1.15–3.47	**0.02**				2.47	1.25–4.88	**0.009**

^§^ additionally adjusted for age; * One woman with missing variable for Last Dentist Visit; statistically significant values in **bold**; BMI – body mass index.

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
