# Peer review of "Periodontal Health and Use of Oral Health Services: A Comparison of Germans and Two Migrant Groups"

_ijerph, 2019, doi:10.3390/ijerph16163000_

Round 1

Reviewer 1 Report

Your research focuses on the periodontal health and use of oral health services among Germans and migrant groups. The topic of this manuscript has the potential to be of foremost importance and interest to the Journal’s readership. However, the content has some major issues and shortcomings that render its value questionable.

Major points:

[1] The tables can confuse the issue. The authors should revise the tables for the readers to understand easily.

[2] Was the assessment of periodontal diseases in this study well known? I think that your assessment may have the overestimation of periodontal disease. In addition, I disagree the sentence “Pocket depth (PD) was used as main indicator for the presence of periodontal inflammation (P3, L119)”. Did you examine the bleeding on probing? The authors should discuss the assessment of periodontal diseases.

Minor

[1]  How many nurses did support in this study and did they calibrate? The results of the calibration?

[2] Manuscript and Tables: Numbers should be rounded off to two significant digits, unless more precision is absolutely necessary.

[3]  P3, L105. Were the numbers of 127 Germans, 67 individuals with a Turkish background, and 56 resettlers correct?  

[4] The abbreviations should be used correctly.

Reviewer 2 Report

1- The sample has 250, not 251 individuals (127 Germans, 67 individuals with a Turkish background, and 56 resettlers)

2- The choice of the variables such as Body Mass Index and Physical activity are not justified

3- In the Results, the authors mentioned that:

(Table 4) Body mass index (0.45), physical activity (0.87), sex (0.02), and household income (0.81) showed no significant association

But in the Discussion the say that: There was a significant association with sex. Males revealed a higher proportion of higher pocket depth than females

4- Regarding the Discussion, the authors mentioned:

4.1- we see also a clear trend in our study population as well, with a strong correlation between age and maximum pocket depth - These results are not showed

4.2- A significant effect of educational level has been observed - The results shown in Table 4: CI 0.35-1.63 P 0.47 are different

The Discussion must be improved

Reviewer 3 Report

Although the analyses include important variabels associated with gum disease, it is missing a few key variables that might be important in an examination of oral health factors. For instance, the inclusion of a stress variable and variables for cardiovascular disease and diabetes, which are associated with gum disease, could possibly help explain some of the association between gum disease and sex and ses.  

Round 2

Reviewer 1 Report

This second version of the paper is a great improvement, the authors are to be commended. However, I think the authors should correct a writing error; "periodontis" (P3L125).

Author Response

Thank you for your kind words. 

The typo has been corrected